# DRIP: Unleashing Diffusion Priors for Joint Foreground and Alpha Prediction in Image Matting

**Xiaodi Li**[1,2], **Zongxin Yang**[3], **Ruijie Quan**[1,2], **Yi Yang**[1,2*]

[1] The State Key Lab of Brain-Machine Intelligence, Zhejiang University, Hangzhou, China
[2] CCAI, College of Computer Science and Technology, Zhejiang University, Hangzhou, China
[3] DBMI, HMS, Harvard University, Boston, USA
`https://github.com/LiNO3Dy/Drip`

## Abstract

Recovering the foreground color and opacity/alpha matte from a single image (*i.e.*, image matting) is a challenging and ill-posed problem where data priors play a critical role in achieving precise results. Due to the limited matting datasets, traditional methods usually struggle to produce high-quality estimation.To address this, we explore the potential of leveraging vision priors embedded in pre-trained latent diffusion models (LDM) for estimating foreground RGBA values in challenging scenarios and rare objects. We introduce `Drip`, a novel approach for image matting that harnesses the rich prior knowledge of LDM models. Our method incorporates a switcher and a cross-domain attention mechanism to extend the original LDM for joint prediction of the foreground color and opacity. This setup facilitates mutual information exchange and ensures high consistency across both modalities. To mitigate the inherent reconstruction errors of the LDM's VAE decoder, we propose a latent transparency decoder to align the RGBA prediction with the input image, thereby reducing discrepancies. Comprehensive experimental results demonstrate that our approach achieves state-of-the-art performance in foreground and alpha predictions and shows remarkable generalizability across various benchmarks.

## 1 Introduction

Image matting aims to isolate the foreground object from composited images, a long-standing and fundamental task in vision intelligence [1]. It is indispensable for various downstream applications, such as media production, virtual reality, and image/video editing [2, 3]. Mathematically, image matting begins with solving the inverse problem of the composition equation:

$$Image_i = \alpha_i \cdot Foreground_i + (1 - \alpha_i) \cdot Background_i, \quad \alpha_i \in [0, 1], \tag{1}$$

where $i$ denotes the index of a pixel. Here, all quantities on the right-hand side are unknown, and the prediction of the alpha matte $\alpha$ and foreground color represents an ill-posed problem.

In the past decade, advances in deep learning have significantly pushed the boundaries of image matting, rapidly becoming the mainstream direction in this field [2, 4–7]. Despite their impressive performance, two challenges remain unresolved in this domain: **(i) high-quality foreground color prediction**. As illustrated in Fig. 1, most matting methods consist of two stages: namely, alpha prediction with neural networks and foreground isolation via post-processing. These methods typically struggle to generalize and recover high-fidelity foregrounds due to the accumulated errors in alpha prediction and post-processing. **(ii) accurate prediction of semi-transparent objects**. When the target to be predicted contains large areas of semi-transparency (*e.g.*, a water glass) or

---

*Corresponding author: Yi Yang.

38th Conference on Neural Information Processing Systems (NeurIPS 2024).

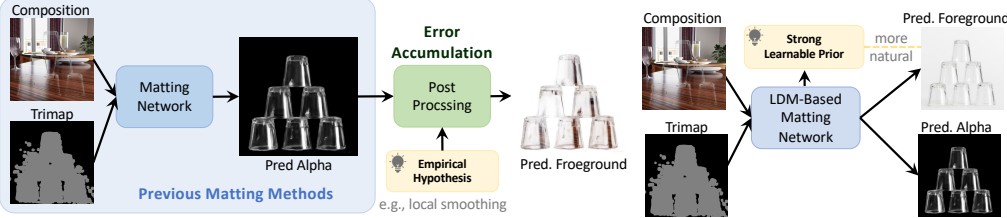

(a) Previous Methods        (b) Drip (ours)

Figure 1: **(a)** Matting methods [20, 4–7] commonly predict the alpha matte and then infer the foreground color by post-processing [21], which often relies on empirical assumptions such as local smoothing of the foreground and background, leading to the accumulation of errors. **(b)** In contrast, our joint prediction approach estimates both the foreground and alpha simultaneously. By leveraging LDM's [8] powerful natural vision prior, our predicted foreground is closer to natural images.

semi-transparent regions with high-frequency details (*e.g.*, patterned semi-transparent fabric), existing methods still struggle to predict the alpha matte accurately.

High-quality semitransparent matting data is difficult to annotate on a large scale. Therefore, *how can we enhance the algorithm's generalization capability for semi-transparent objects and achieve high-quality foreground prediction with limited training data?* Recently, with the emergence of large pre-trained generative models [8–12], their data priors learned from billions of images (*e.g.*, LAION [13]) are found to be useful for various downstream tasks [14–18]. Back to the challenging image matting, we posit that the data priors learned from tons of natural images are also intuitively beneficial. Hence, our key insight is to unleash the data priors from the large pre-trained generative models (LDM) to estimate alpha and foreground simultaneously.

To achieve this, we propose `Drip`, the unleashing Diffusion priors for joint foreground and alpha prediction method, which follows the diffusion paradigm and jointly generates the foreground and alpha map conditioned on the image and trimap input. Specifically, we wisely design a **cross-domain switcher** that leverages domain-aware embedding to unify the foreground and alpha generations in a single-diffusion model. This design facilitates mutual information exchange and ensures high consistency between foreground image and alpha. Besides, the pre-trained VAE compresses the image into a compact latent space, significantly reducing training consumption while inevitably missing detailed information. To narrow the errors caused by VAE, we introduce an auxiliary **latent transparency decoder**, which is implemented by inserting the features from early layers in the encoder into the decoder with several learnable zero-conv layers [19]. This latent transparency decoder significantly contributes to high-fidelity foreground image and alpha prediction and also effectively adapts the pre-trained LDM into image matting.

We extensively evaluate the performance of our method through extensive experiments and comparisons. The results demonstrate that our approach achieves state-of-the-art performance on the Composition-1k test set and exhibits stronger generalizability on other benchmark datasets. Remarkably, `Drip` outperforms all the previous methods in the mainstream benchmark, Composition-1k, where `Drip` improves the SAD metric of alpha prediction by **3.3**% and foreground by **12.1**% and MSE metric of alpha by **6.1**% and foreground by **28.33**%. In summary, the key contributions of this paper are as follows.

- To our best knowledge, we introduce the first LDM-driven matting method, `Drip`, which effectively unleashes the data priors learned from LDM into image matting.
- To enable joint prediction of foreground and alpha, we propose a switcher and a cross-domain attention mechanism, facilitating mutual information exchange and ensuring high consistency.
- To mitigate the inherent reconstruction errors of the LDM's VAE decoder, we propose a latent transparency decoder to align the RGBA prediction with the input image.

## 2  Related Work

**Image Matting** is aimed to extract the foreground objects from arbitrary natural images [22, 2]. Traditional methods always need the auxiliary user input like trimap [23, 4] and scribble [24, 25]. These methods basically only leverage low-level color or structure features, which limits their ability

to distinguish foreground details from images. With the success of deep learning, researchers have begun to use deep convolutional neural networks (CNNs) to predict the alpha map in an end-to-end fashion [26, 20]. One type takes images and auxiliary trimap or scribble as input and outputs the alpha map [5]. In order to alleviate the demand for trimap, trimap-free methods [27, 6, 28] are proposed to directly predict alpha mattes from the input image, which increases efficiency while sacrificing performance. Although these existing methods achieve impressive results in alpha prediction, accurately predicting foreground and background colors remains an essential yet challenging task for high-quality matting. Tang *et al.* [29] and Aksoy *et al.* [30] firstly proposed to address color estimation by sequentially or directly predicting the background and foreground colors before alpha prediction. Furthermore, recent method [31] unifies foreground, background, and alpha matte into an end-to-end framework. While, another line of works [32, 28] focuses on foreground human extraction and alpha matte prediction. However, these methods are limited by the lack of high-quality labeled data. Meanwhile, the explosion of generative models shows immense potential in providing priors in different tasks. In this work, we explore unleashing diffusion priors within stable diffusion [8] to improve the performance of image matting.

**Diffusion Models** have emerged as a powerful class of generative models, which learn a reverse denoised process from the Gaussian noise to natural images [33]. In the vanilla DDPMs [33], the sampling process is time-consuming due to the Markovian property. To speed up the sampling, DDIMs [34] is proposed to provide a non-Markovian shortcut. Furthermore, LCM [35] just formulate the diffusion process as one-step denoising via an ODE. Besides the speed, a series of works [36, 10, 37, 38] focus on increasing the controllability of diffusion models. For instance, Controlnet [19] fine-tunes a Stable Diffusion model with zero convolutions, which proves to be effective in adapting the pre-trained diffusion models to different tasks by adding different conditions.

**Diffusion Priors in Visual Perceptive Tasks** are prevalent and hot topics. A series of works leverage the diffusion priors in segmentation [17], image enhancement [39], depth estimation [40] and 3D vision [41, 42]. In the context of image matting, Xu *et al.* [43] propose to formulate alpha prediction as a denoised process, and train a condition generation model in DDPMs fashion. However, the vanilla DDPMs have not been scaled-up training due to their expensive computational cost. On the contrary, LDM [8] proposes to compress the features into a compact latent space, which obviously reduces the computational cost. And based on it, Stable Diffusion is largely trained on the large-scale dataset [13]. However, LDM generates the image features in the latent space encoded by pretrained VAE. In order to alleviate the domain gap, Marigold [16] finetunes UNet backbone of diffusion models to perform affine invariant monocular depth estimation and exhibit strong generalization capability. Inspired by this, we carefully discuss and propose a novel method to unleash the diffusion priors within stable diffusion to improve the performance of image matting while preserving high-fidelity details.

## 3 Drip

**Overview**. Drip is an LDM-based matting model designed to predict both foreground and alpha values while ensuring high consistency between these two representations. Given an input image $(X)$ and a trimap $(Tri)$ indicating the object to be matted, our goal is to estimate its corresponding Foreground $(F)$ and alpha $(\alpha)$. Initially, we explore the problem using the diffusion paradigm (see Sec. §3.1). Subsequently, we present our LDM-based matting model (see Sec. §3.2). This model employs a cross-domain switcher to simultaneously generate the foreground color and alpha map using a single diffusion model. Moreover, through mutual information exchange, the model effectively enhances boundary and texture consistency. To address the challenge of missing high-frequency information caused by VAE compression, the model incorporates an auxiliary latent transparency decoder (see Sec. §3.3). An overview of Drip is provided in Fig. 2.

### 3.1 Problem Formulation

The task of foreground and alpha estimation is to model the mapping $f(\cdot) : (X, Tri) \rightarrow (F, \alpha)$, where $F \in R^{H \times W \times 3}$ represents the foreground and $\alpha \in R^{H \times W}$ represents the alpha map. The input conditions are an RGB image $X \in R^{H \times W \times 3}$ and a trimap $Tri \in R^{H \times W}$, which consists of three values indicating the foreground, unknown, and background regions, respectively. However, unlike prior works that adopt CNN or transformer as architecture, we employ a diffusion-based scheme $f(\cdot)$ to model the joint foreground and alpha distribution $p(F, \alpha)$.

Diffusion Probabilistic Models [44, 33] define a forward Markov chain that progressively transits the sample $x$ drawn from data distribution $p(x)$ into noisy versions $x_t \in (1, T) | x_t = \alpha_t x_0 + \sigma_t \epsilon$, where

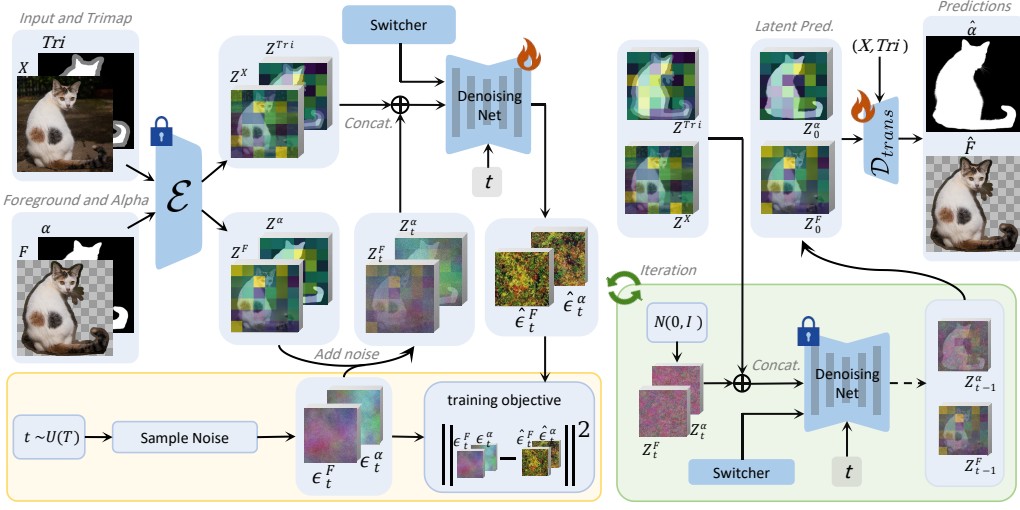

(a) Training        (b) Inference

Figure 2: **Overview of** `Drip`. **(a)** During training, the input image $X$, trimap $Tri$, ground-truth foreground $F$, and ground-truth alpha map $\alpha$ are first encoded into latent representations $Z^X$, $Z^{Tri}$, $Z^F$, and $Z^\alpha$ respectively using the original Stable Diffusion VAE encoder $\mathcal{E}$. After adding noise to $Z^F$ and $Z^\alpha$, all the latents are fed into a U-Net, which generates the output in the foreground or alpha domain guided by a switcher (§3.2.1). The U-Net is then fine-tuned by optimizing the standard diffusion objective(§3.2.3). **(b)** After executing the T-step denoising schedule, the resulting latents $Z_0^F$ and $Z_0^\alpha$ are decoded by a transparent latent decoder (§3.3).

$\epsilon \sim N(0, I)$, $T$ is the timestep, $\alpha_t$ and $\sigma_t$ are the noisy scheduler terms that control sample quality. In the reverse Markov chain, it learns a denoising network $\epsilon_\theta(\cdot)$ parameterized by $\epsilon$ usually structured as U-Net [45] to transform $x_t$ into $x_{t-1}$ from an initial Gaussian sample $x_T$ through iterative denoising.

For the joint foreground and alpha distribution $p(F, \alpha)$, given a conditional input image $X$ with its corresponding trimap $Tri$, the foreground $F$ and the alpha map $\alpha$ can be obtained by the generative formulation in Markov probabilistic form:

$$f(X, Tri) = p\left(\hat{F}_T, \hat{\alpha_T}\right) \prod_{t=1}^{T} p_\theta(\hat{F_{t-1}}, \hat{\alpha_{t-1}}|\hat{F}_t, \hat{\alpha}_t, X, Tri), \quad \hat{F}_T, \hat{\alpha_T} \sim N(0, I). \quad (2)$$

To enhance computational efficiency and generate higher-resolution images, Stable Diffusion [8] employs the latent diffusion model, where the diffusion steps are performed in the low-dimensional latent space instead of directly operating on the original data. The latent space is formed within the bottleneck of VAE [46], which is trained separately from the denoiser. This design allows latent space compression and facilitates perceptual alignment with the data space.

To translate our formulation (Eq. 2) into the latent space, we obtain the corresponding latent code for a given image using an encoder: $z^{(i)} = \mathcal{E}(i)$, where $i \in X, Tri, F, \alpha$. It's worth noting that we triplicate the single-channel trimap and alpha map into three channels. Moreover, the denoiser $\epsilon_\theta(\cdot)$ is subsequently trained in the latent space. To obtain the desired outputs, given latent codes $z_F$ and $z_\alpha$, the foreground and alpha can be reconstructed using the decoder $\mathcal{D}$: $\hat{F} = \mathcal{D}(z^F)$ and $\hat{\alpha} = \mathcal{D}(z^\alpha)$. It is worth noting that in the Matting task if the reconstructed image is obtained directly from the latent representation of the foreground and alpha without making any modifications to the VAE, a significant error can occur.

## 3.2 LDM-Based Matting Model

We base our model on a pretrained text-to-image LDM (Stable Diffusion v2 [8]), which has learned strong and generalizable image priors from LAION-5B [13]. In order to accept a given image and trimap as conditions and simultaneously generate both foreground and alpha outputs, we quadruple the input of the original U-Net and employ a switcher mechanism to expand the capabilities of the original LDM model. Additionally, we incorporate cross-domain attention to enhance consistency.

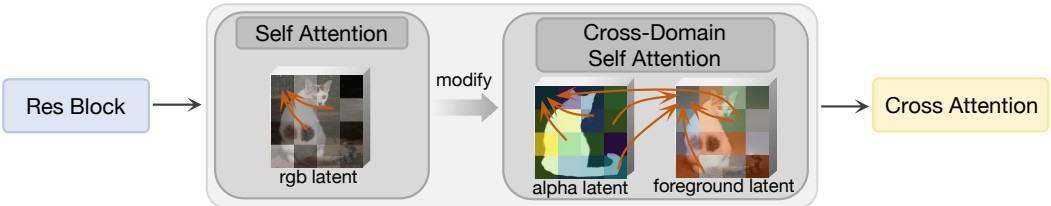

Figure 3: **Demonstration of Cross-Domain Attention**(§3.2.2). To enhance mutual guidance and ensure contextual consistency, we instead utilize a cross-domain self-attention mechanism instead of self-attention to associate the foreground and alpha latent.

### 3.2.1 Foreground and Alpha Switcher

Previous matting models [20, 4–6] have primarily focused on predicting the alpha value and utilizing post-processing methods, such as the local smoothness assumption, to estimate the foreground color [21, 47]. However, this two-stage approach often leads to error accumulation and suboptimal foreground estimation results, particularly when dealing with transparent objects. To address this limitation, we propose a novel approach that leverages an LDM-based method to predict the foreground and alpha values simultaneously. Our method generates more realistic foreground images by incorporating a strong natural image prior distribution learned from Stable Diffusion v2 [8].

To incorporate foreground and alpha estimation, one straightforward approach is to finetune two U-Nets separately to model their respective distributions. However, this method introduces additional parameters and fails to capture the inherent connections between foreground and alpha. Motivated by the work [48, 14], we propose a novel approach using a switcher that enables a single stable diffusion model to generate both foreground and alpha values based on indicators. Mathematically, the foreground and alpha values can be obtained as follows:

$$\hat{F} = f(X, Tri, s_F) = f(X, Tri, \text{PosEnc}(1)) \tag{3}$$

$$\hat{\alpha} = f(X, Tri, s_\alpha) = f(X, Tri, \text{PosEnc}(0)) \tag{4}$$

In the above equations, $s_F$ and $s_\alpha$ are one-dimensional vectors controlling the foreground and alpha domains, respectively. The switchers are encoded using low-dimensional positional encoding and combined with time embedding within the U-Net architecture.

### 3.2.2 Cross-Domain Attention

To further facilitate mutual-guided optimization, we introduce a modification to the self-attention layer in the U-Net architecture, transforming it into a cross-domain self-attention layer that encourages spatial alignment (refer to Fig 3). This operator enhances the geometric consistency between the foreground and alpha channels and accelerates convergence. The cross-domain attention operation, denoted as $AttCD(\cdot)$, is defined as follows:

$$AttCD(Q_i, K_i, V_i) = Att\left(W_q \cdot h_i, W_k \cdot (h_F \oplus h_\alpha), W_v \cdot (h_F \oplus h_\alpha)\right) \tag{5}$$

Here, $i = F, \alpha$, $h_F$ and $h_\alpha$ represent the latent embeddings of the foreground and alpha channels within the transformer blocks, respectively. The symbol $\oplus$ denotes the concatenation operation. $W_q$, $W_k$, and $W_v$ are the matrices of query, key, and value embeddings, respectively. Finally, $Att(\cdot)$ refers to the softmax attention mechanism.

### 3.2.3 LDM Loss Function

We adopt annealed multi-resolution noise noises [16] to preserve low-frequency details in the depth and normal maps, as similar values will frequently appear in local geometric regions. This deviation proves to be more efficient than a single-scale noise schedule. We perturb the two geometry branches with the same timestep scheduler to decrease the difficulty when learning more modalities. The canonical standard learning objective we utilize is defined as follows:

$$\mathcal{L} = \mathbb{E}_{X,Tri,F,\alpha,\epsilon,t}[\epsilon_\theta\left(F_t; X, Tri, s_F\right) - \epsilon_t^F \|_2^2 + \epsilon_\theta\left(\alpha_t; X, Tri, s_\alpha\right) - \epsilon_t^\alpha \|_2^2] \tag{6}$$

Here, $\epsilon_t^F$ and $\epsilon_t^\alpha$ are two Gaussian noises independently sampled from annealed multi-scale noise sets for the foreground and alpha, respectively.

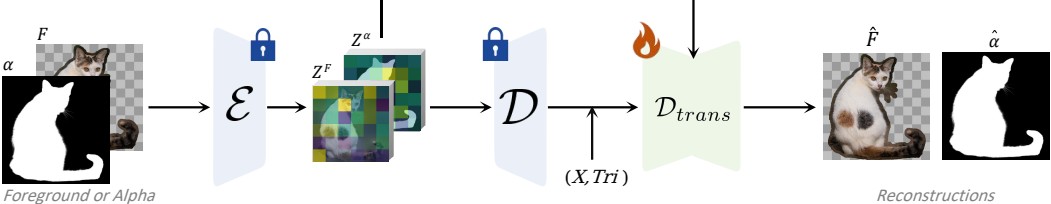

Figure 4: **Structure of Transparent Latent Decoder** (§3.3). Due to the non-negligible reconstruction loss introduced by the compression of the LDM's VAE, we employ a transparent latent decoder, which takes the output images of the LDM-based matting model and the corresponding latent as inputs, generating results that are more consistent with the details of the composite image.

## 3.3  Latent Transparency Decoder

The compression of the VAE [46] in LDM introduces a non-negligible reconstruction loss, indicating that $x \neq \mathcal{D}(\mathcal{E}(x))$. Previous discriminative tasks using the LDM [8, 12] have primarily focused on tasks such as segmentation and depth estimation. In comparison, matting is a task that places more emphasis on capturing fine details, making the errors introduced by the VAE more noticeable.

To address this challenge, we draw inspiration from LayerDiffusion [49] and propose a novel latent transparency decoder $\mathcal{D}_{trans}$. This decoder takes the output images of the LDM-based matting model 3.2 and the corresponding latent as inputs to generate results that are more consistent with the details of the composition image (refer to Fig 4). Mathematically, the expression is given by

$$(\hat{F}, \hat{\alpha}) = \mathcal{D}_{trans}(X, Tri, \mathcal{D}(\hat{z^F}), \mathcal{D}(\hat{z^\alpha}), \hat{z^F}, \hat{z^\alpha}) \tag{7}$$

where $\hat{z^F}$ and $\hat{z^\alpha}$ represent the foreground and alpha predictions in the latent space, respectively. Additionally, $\mathcal{D}$ represents the original VAE.

By incorporating this transparent latent decoder into our framework and training it with the matting loss, we aim to improve the fidelity of the predicted foreground and alpha outputs, ensuring that they capture the intricate details present in the composition image. This enhancement is particularly crucial for the task of matting, which relies heavily on preserving fine details and boundaries.

## 4  Experiment

### 4.1  Experimental Setup

#### 4.1.1  Datasets.

**Composition-1k dataset** [5] is a synthetic dataset consisting of 431 manually labeled foreground images for training and an additional 50 foreground images for evaluation. The training set is created by compositing each foreground image with 100 background images sourced from the COCO dataset [50]. This approach allows for the generation of a sufficient training set despite having a smaller number of unique foreground object images. Similarly, the test set is generated by synthesizing 50 test foreground images using 20 background images from VOC2012 [51], resulting in a total of 1000 test images.

**AIM-500 dataset** [6] is a benchmark for natural image matting that encompasses various object categories. It consists of 500 high-resolution real nature images, each with a minimum short side length of 1080 pixels. Unlike other natural matte datasets [52–54] that are often limited to specific classes of human and animal images, AIM-500 offers a more diverse range of objects. Evaluating the performance on the AIM benchmark helps assess the model's ability to generalize well to natural images rather than solely fitting the distribution of synthetic images. Therefore, the evaluations conducted on both the Composition-1k and AIM benchmarks complement each other, providing a comprehensive understanding of matte models in real-world scenarios.

#### 4.1.2  Implementation Details.

We employed Diffusers [55] with Stable Diffusion v2 [8] as the backbone for implementing our Drip. Text conditioning was disabled by providing empty text input. To accommodate the two additional conditions, the weights of the first layer of the UNet were copied three times as initialization. After training the LDM-based matting network, the obtained output and latent results were utilized as inputs for training the latent transparent decoder.

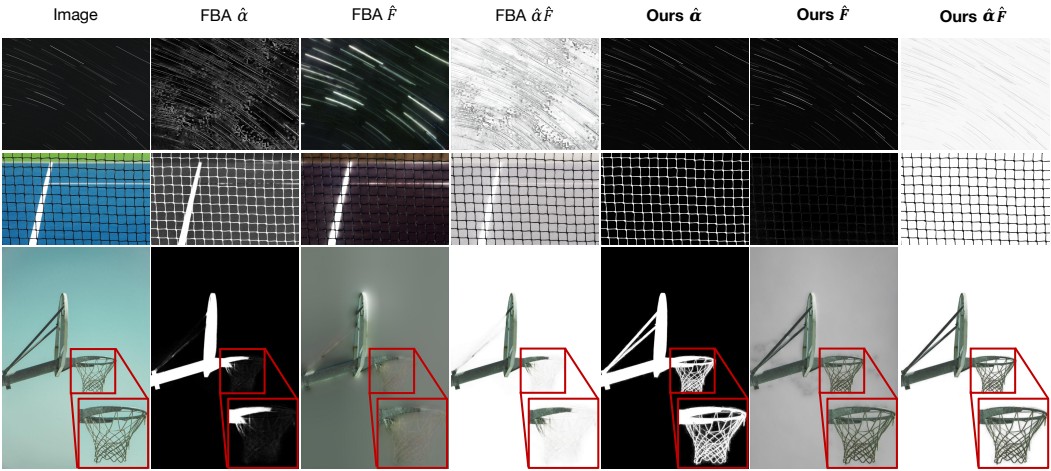

Figure 5: **Qualitative Result of Foreground.** The visual results compared with FBA [31] on AIM-500[6]. Please zoom in for the best view.

Table 1: Comparison of our Drip with State-of-the-Arts (SOTAs) on the synthetic dataset Composition-1k [5] and the natural dataset AIM-500 [6], focusing on four metrics for alpha prediction results.(§4.2).

| | Method | Publication | Composition-1k | | | | AIM-500 | | | |
|---|---|---|---|---|---|---|---|---|---|---|
| | | | SAD | MSE | Grad | Conn | SAD | MSE | Grad | Conn |
| Alpha | DIM [5] | CVPR'17 | 50.4 | 14.0 | 31.0 | 50.8 | 49.3 | 14.7 | 29.3 | 47.1 |
| | IndexNet [56] | ICCV'19 | 45.8 | 13.0 | 25.9 | 43.7 | - | - | - | - |
| | FBA [31] | ArXiv'20 | 25.8 | 5.2 | 10.6 | 20.8 | - | - | - | - |
| | HATT [27] | CVPR'20 | 44.0 | 7.0 | 29.3 | 46.4 | 479.2 | 270.0 | 238.6 | 474.0 |
| | AIM [6] | IJCAI'21 | - | - | - | - | 43.9 | 16.1 | 33.1 | 43.2 |
| | GFM [32] | IJCV'22 | - | - | - | - | 52.7 | 21.3 | 46.1 | 52.9 |
| | MFormer [57] | CVPR'22 | 23.8 | 4.0 | 8.7 | 18.9 | - | - | - | - |
| | ViTMatte [7] | IF'23 | 21.5 | 3.3 | 7.2 | **16.2** | - | - | - | - |
| | DiffMat [43] | ArXiv'24 | 22.8 | 4.0 | 6.8 | 18.4 | - | - | - | - |
| | Ours | - | **20.8** | **3.1** | **6.8** | 17.8 | **17.3** | **1.5** | **5.4** | **14.7** |

To enhance the diversity of the dataset, we performed various data augmentation techniques during the training of the 2D image set. These included random horizontal flipping, cropping, and photometric distortion. Besides, in order to enhance the foreground, we followed LayerDiffusion [49] to fill the foreground image. Pixels with Alpha values equal to zero in the foreground have no impact on the appearance of the alpha-blended image when assigning any color. Nevertheless, since neural networks tend to produce high-frequency patterns surrounding image edges, avoiding unnecessary edges in the RGB channels prevents potential artifacts. To achieve this, we applied Gaussian blurring to regions of the foreground image where the Alpha value was strictly equal to zero.

During training, the DDPM noise scheduler [58] with 1000 diffusion steps was applied. At inference time, the DPM solver scheduler [59]was employed, and only 10 steps were sampled. The model was trained for 30,000 steps with a total batch size of 96, using an image size of $512 \times 512$ exclusively on the Composition-1k dataset [5]. The entire training procedure typically took approximately 2 days when executed on a cluster consisting of 4 Nvidia Tesla A100-80GB GPUs. For optimization, the Adam optimizer was used with a learning rate of $1 \cdot 10^{-5}$.

### 4.1.3 Evaluation Metric.

We employ standard evaluation metrics to assess the quality of alpha predictions. Specifically, we report the Sum of Absolute Differences (SAD), Mean Square Error (MSE), Gradient loss (Grad), and Connectivity loss (Conn). A lower value for these metrics indicates a higher quality alpha matte.

Additionally, we follow the evaluation methodology of FBA matting [31] to evaluate the quality of foreground predictions. Since colors other than the foreground object region are not used, we only consider the region where the ground truth alpha, denoted as $\alpha_{gt}$, is located and use it as a weighting factor. We apply the SAD and MSE to evaluate the $\alpha_{gt}F$ predictions. A lower value for these metrics indicates a better quality foreground prediction.

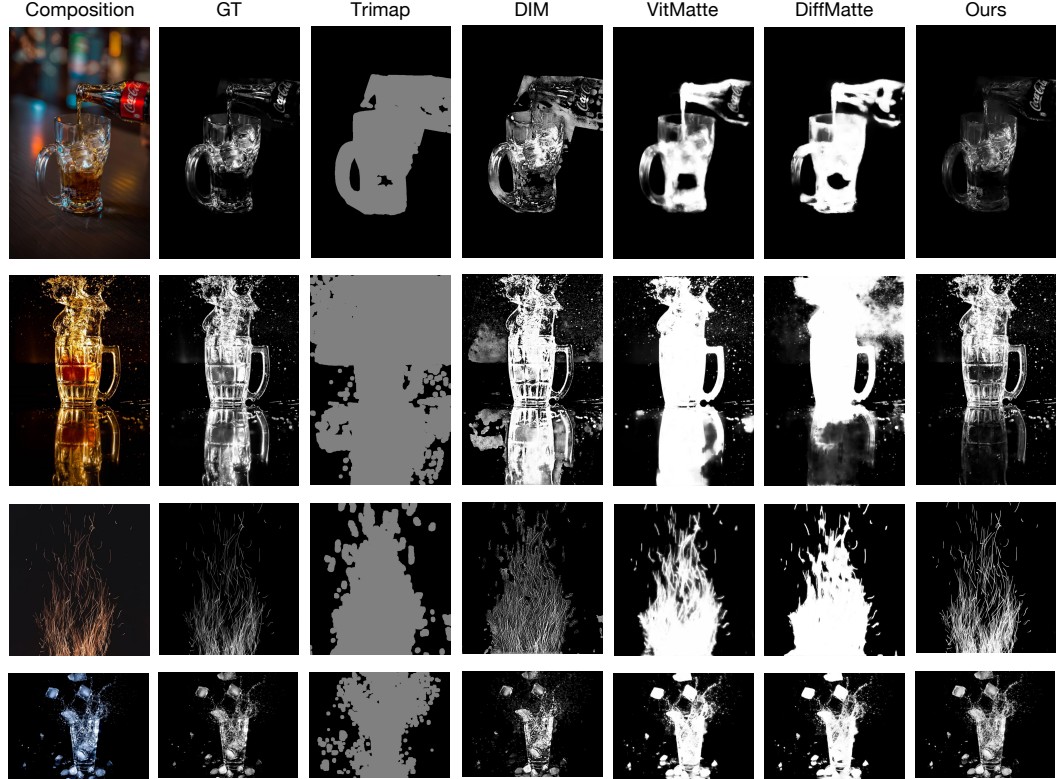

| Composition | GT | Trimap | DIM | VitMatte | DiffMatte | Ours |

Figure 6: **Qualitative Result of Alpha.** The visual results compared with previous SOTA methods on AIM-500[6]. Please zoom in for the best view.

### 4.1.4 Baselines.

For **alpha estimation**, we consider several state-of-the-art methods as baselines, including both trimap-based approaches and automated methods. Among the automated methods, HATT [27], AIM [6], and GFM [32] are included. On the other hand, the trimap-based methods consist of DIM [5], IndexNet [56], FBA [31], MFormer [57], and ViTMatte [7]. These trimap-based methods are all deep learning-based, with the backbone architecture gradually transitioning from convolutional networks to transformer-based networks. Furthermore, with the development of diffusion, the contemporary method DiffMat [43] is also based on diffusion. However, it is important to note that DiffMat [43] does not operate in the latent space, nor does it leverage the priori of the natural image distribution learned from diffusion.

For **foreground estimation**, we compare our proposed method with a limited set of baselines, as many matting algorithms primarily focus on alpha estimation rather than foreground estimation. In our comparison, we include Global-Matting [60] and KNN-Matting [61] as non-deep learning methods. We consider ContextAware-Matting [62] and FBAMatting [31] for deep learning-based methods. These approaches leverage deep neural networks to estimate the foreground and have demonstrated promising results in previous studies.

### 4.2 Quantitative Result & Qualitative Result

The results are shown in Table 1 for alpha. The results indicate that our method outperforms others by a large margin and achieves state-of-the-art (SOTA) performance. On the Composition-1k dataset, our method improves the SAD metric by 0.7 (+3.3%) and the MSE metric by 0.2 (+6.1%) compared to the ViTMatte [7] method. The performance improve-

Table 2: Comparison with SOTAs(§4.2).

| | Method | | Publication | Composition-1k | |
| | | | | SAD | MSE |
|---|---|---|---|---|---|
| Foreground | Global | [60] | CVPR'11 | 220.39 | 36.3 |
| | KNN | [61] | TPAMI'13 | 281.9 | 36.3 |
| | CA | [62] | ICCV'19 | 61.72 | 3.24 |
| | LBS | [29] | CVPR'19 | 49.7 | 8.6 |
| | FBA | [31] | ArXiv'20 | 38.8 | 6.0 |
| | Ours | | - | **34.1** | **4.3** |

ments are even more noticeable on the AIM-500 dataset, where our method improves the SAD by 26.6 and the MSE by 14.6 compared to the AIM [6] approach. For the foreground metrics, as shown in Table 2, our method demonstrates significant improvements compared to the FBA-Matting [31]. Specifically, we improve the SAD metric by 4.7 (+12.1%) and the MSE metric by 1.7 (+28.33%).

Table 3: A set of ablative experiments about our proposed modules on the AIM-500 [6].(§4.3)

| Switcher | CDAttn | $\hat{\alpha}$ | | $\alpha\hat{F}$ | | $\hat{\alpha}\hat{F}$ | |
|---|---|---|---|---|---|---|---|
| | | SAD | MSE | SAD | MSE | SAD | MSE |
| ✓ | | 18.1 | 1.7 | 21.1 | 3.8 | 23.7 | 5.2 |
| | ✓ | 17.8 | 1.5 | 21.3 | 4.1 | 26.7 | 5.9 |
| ✓ | ✓ | 17.3 | 1.5 | 20.6 | 3.7 | 23.3 | 4.9 |

| Decoder | $\hat{\alpha}$ | | $\alpha\hat{F}$ | |
|---|---|---|---|---|
| | SAD | MSE | SAD | MSE |
| | 21.3 | 2.1 | 25.3 | 4.8 |
| ✓ | 17.3 | 1.5 | 20.6 | 3.7 |

(a) Joint prediction of foreground and alpha

(b) Transparent latent decoder

Additionally, we visualize some qualitative results in comparison with other baselines. As demonstrated in Figure Fig. 5, our method produces foreground predictions that are more closely aligned with natural images, and the resulting RGBA outputs are more consistent with the original image in detail. As shown in Fig. 6, for natural datasets, our results are significantly closer to the true values, demonstrating the strong generalization ability of our method.

## 4.3 Ablation Study

We conduct ablation studies on AIM-500 [6] to investigate the contributions of the proposed joint prediction of foreground and alpha and the transparent latent decoder.

**Joint Prediction of Foreground and Alpha** We first investigate the effect of the proposed cross-domain attention mechanism on the joint foreground and alpha estimation(*cf.* §3.2). As shown in Table 3a, when the cross-domain attention module is removed, we observe a decrease in the prediction accuracies of both the foreground and alpha representations. Importantly, the accuracy of the final RGBA composition, obtained by multiplying the predicted foreground and alpha, decreases more significantly. This suggests that the information interaction between the two modalities, facilitated by cross-domain attention, increases the consistency of spatial information between the foreground and alpha representations. This finding verifies that cross-domain self-attention can effectively correlate the two representations, enabling them to benefit from each other's contextual cues mutually.

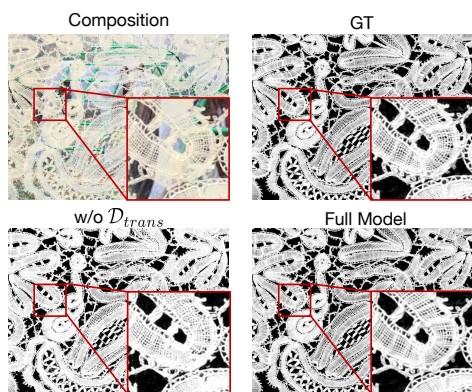

Figure 7: **Ablation Result.** When the transparent latent decoder is not used, the generated output exhibits significant differences in low-level details with the original input.

For the switcher, we also notice a clear reduction in performance, suggesting that giving additional embedding information rather than simply distinguishing between foreground and alpha with channel order is helpful in relation to the neural network's ability to distinguish and utilize information from the two modalities.

**Transparent Latent Decoder** Next, we ablate the effect of the $\mathcal{D}_{trans}$(*cf.* §3.3). As reported in Table 3b, using the original VAE decoder without $\mathcal{D}trans$ leads to a degradation in the prediction performance of both alpha and foreground. In addition, the qualitative results on composition-1k also clearly show that the low-level details correspond better after applying $\mathcal{D}_{trans}$, mitigating the errors induced by VAE compression.

## 5 Conclusion and Future Work

This work addresses two key challenges in image matting: high-quality foreground prediction and accurate semi-transparent object alpha estimation. To overcome these, we propose Drip method, which leverages data priors from large pre-trained generative models to jointly predict the foreground and alpha. Drip utilizes a switcher and a cross domain attention for consistent foreground-alpha generation and a latent transparency decoder to enhance fidelity. Extensive experiments demonstrate Drip achieves sota performance on Composition-1k and stronger generalization on other benchmarks.

While the Drip method demonstrates strong performance in image matting tasks, several key limitations warrant consideration: **i) Model Complexity and Deployment**. The incorporation of latent diffusion models (LDMs) substantially increases the architectural complexity of the approach. This added complexity may impact deployment and inference efficiency, particularly in real-time or

resource-constrained environments where computational overhead is critical. **ii) Inherited Biases from Generative Priors**. Drip's methodology relies on the extensive priors captured within pretrained LDMs. Consequently, Drip inherits any biases present in the original generative model. These biases could adversely affect the method's performance on certain types of images or domains, such as those with highly complex lighting and shadows, or those containing numerous small texture details, thereby limiting its general applicability.

**Acknowledgements.** This work was supported by the National Natural Science Foundation of China (U2336212) and the Fundamental Research Funds for the Central Universities (No. 226-2022-00051).

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

# A  Appendix

This appendix contains additional details for the Neurips 2024 submission, titled *"DRIP: Unleashing Diffusion Priors for Joint Foreground and Alpha Prediction in Image Matting"*. The appendix is organized as follows:

- §A.1 offers more implementation details of the multi-level annealed noise.

- §A.2 offers more implementation details of the augmentation to pad the foreground.

- §A.3 provides more information about the transparent latent decoder.

- §A.4 provides ablation results on the number of timesteps for the AIM [6] dataset.

## A.1  Annealed Multi-Resolution Noise

In order to make ldm handle the details better, we adopt the annealed multi-resolution noise used in [16, 14] .In crafting the standard multi-resolution noise, an array of Gaussian noise images is initially sampled to construct a pyramid with a gradient of resolutions, which are then merged using upscaling, weighted averaging, and normalization methods. Each level $i$ of this pyramid is assigned a weight following the formula $s^i$, where $s$ is a decimal fraction between 0 and 1, signifying the extent of the influence exerted by noise at reduced resolutions. To better align the resultant noise with the Gaussian distribution outlined in the foundational DDPM framework, the weights of the upper levels $i > 0$ are modified in accordance with a diffusion schedule. Specifically, at each time step $t$, the $i$-th level is endowed with a weight calculated as $(s^i/T)^t$, with $T$ representing the cumulative number of diffusion steps. As a result, the weight allocated to levels with diminished resolution is progressively reduced as the schedule nears its noise-free terminal point.

---

**Algorithm 1** annealed_pyramid_noise(x,timesteps,discount)

---

1: $b, c, w, h \leftarrow x.shape$ {Get the shape of the input tensor}
2: $w_{ori} \leftarrow w, h_{ori} \leftarrow h$ {Save the original dimensions}
3: $noise \leftarrow gen\_noise\_like(b * c * w_{ori} * h_{ori})$ {Create a noise tensor}
4: $i \leftarrow 0$
5: **repeat**
6:     $r \leftarrow rand() * 2 + 2$ {Generate a random scale factor}
7:     $w \leftarrow \max(1, \lfloor w_{ori}/r^i \rfloor)$
8:     $h \leftarrow \max(1, \lfloor h_{ori}/r^i \rfloor)$ {Compute the current feature map size}
9:     $temp\_noise \leftarrow gen\_noise\_like(b * c * w * h)$ {Generate a temporary noise tensor}
10:    **for** $j \leftarrow 0$ to $b * c * w_{ori} * h_{ori} - 1$ **do**
11:        $x_{idx} \leftarrow j\%w_{ori}$
12:        $y_{idx} \leftarrow \lfloor j/w_{ori} \rfloor$
13:        $new\_x \leftarrow \lfloor x_{idx} * (w_{ori}/w) \rfloor$
14:        $new\_y \leftarrow \lfloor y_{idx} * (h_{ori}/h) \rfloor$
15:        $new\_idx \leftarrow new\_y * w + new\_x$
16:        $noise[j] \leftarrow noise[j] + temp\_noise[new\_idx] * (timesteps/1000.0) * discount^i$
17:    **end for**
18:    $i \leftarrow i + 1$
19: **until** $i \geq 10$ or $(w \leq 1$ and $h \leq 1)$ {If already reached minimum resolution, break out}
20: $result \leftarrow noise/noise.std$ {Normalize the noise tensor}
21: **return** $result$

---

## A.2  Padded Foreground

In order to enhance the foreground effect and avoid potential artifacts, we apply a fill technique [49] to the foreground image. Pixels in the foreground with an alpha value equal to zero, regardless of their color, do not affect the appearance of the final alpha-blended image. However, neural networks tend to produce high-frequency patterns at the edges of the image, so avoiding unwanted edges in the RGB channel helps prevent potential artifacts. To address this issue, we apply Gaussian blurring to regions of the foreground image where the alpha value is strictly equal to zero.

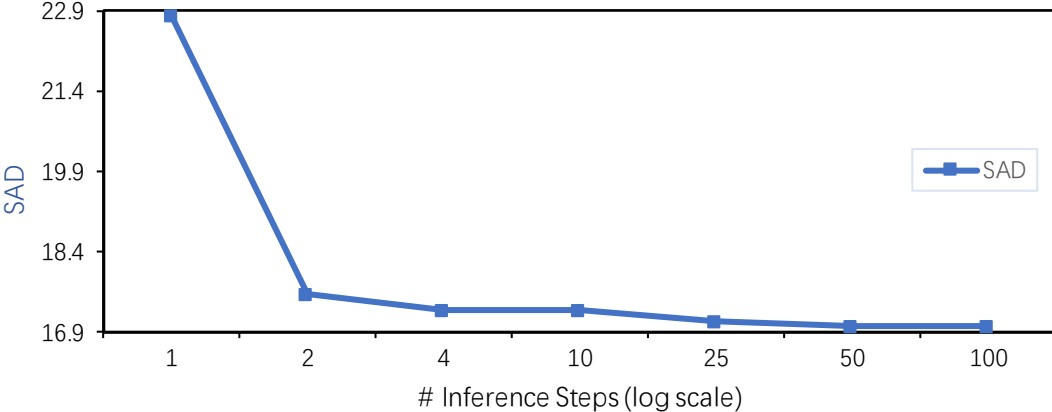

Figure 8: **Impact of Denoising Steps on Performance.** The performance of our method improves as the number of denoising steps increases, with diminishing marginal gains as the number of timesteps becomes larger.

---

**Algorithm 2** padded_fg(alpha, fg)

---

$h, w \leftarrow$ shape($alpha$) {Get the height and width of the input image}
2: mask $\leftarrow$ gen_mask($alpha$) {a mask where alpha = 0 are 1 and alpha > 0 are 0}
$i \leftarrow 0$
4: **while** $i < 64$ **do**
filtered_image $\leftarrow$ gaussian_blur($fg, (13, 13), 0$) {apply gaussian blur to foreground}
6: $j \leftarrow 0$
**while** $j < h \times w$ **do**
8: **if** mask[$j$] == 1 **then**
$fg[j] \leftarrow$ filtered_image[$j$] {update foreground if mask indicates transparent region}
10: **end if**
$j \leftarrow j + 1$
12: **end while**
$i \leftarrow i + 1$
14: **end while**
**return** $fg$ {Return the updated foreground image}

---

### A.3 Detail of Transparent Latent Decoder

The core component of our approach is the transparent latent decoder, which is implemented as a U-Net architecture. To train this module, we randomly sample the step number between 1 and 10 to get the output of latent matting network, which we then use as input. We utilize a loss function commonly used in matting algorithms to optimize the model. Specifically, we employ the combined matting loss, which includes separate components such as the $l_1$ loss [7, 63], $l_2$ loss [63], laplacian loss [62, 63], and gradient penalty loss [7] for both alpha and foreground. The objective function becomes:

$$L_{mat} = L_{l_1}^{sp} + L_{l_2} + L_{lap} + L_{grad} \tag{8}$$

### A.4 Ablation Results on Timesteps

We conducted ablation studies on the number of diffusion timesteps used in our method on the AIM [6] dataset. As shown in Figure 8, increasing the number of timesteps generally improves the performance, though the gains diminish as the number of timesteps becomes larger.

