# OpenReview forum: "DRIP: Unleashing Diffusion Priors for Joint Foreground and Alpha Prediction in Image Matting"
_NeurIPS.cc/2024/Conference — NeurIPS 2024 poster_

### Official Review · Reviewer_TjXY · 2024-07-04

**Soundness:** 3
**Presentation:** 3
**Contribution:** 2
**Rating:** 5
**Confidence:** 4

**Summary:**

The paper describes a new alpha matting model, which builds upon Stable Diffusion v2, adding various blocks including a switcher for training for alpha and foreground color prediction using the same model, cross-domain attention and an alpha decoder. Unlike the usual emphasis on optimizing models for solely alpha prediction, the authors also tune their model to also accurately predict foreground colors. Their results are competitive with the state-of-the-art alpha matting models in alpha estimation.

**Strengths:**

* The emphasis on estimating foreground colors is spot on. In fact alpha mattes by themselves are of little practical use in the absence of foreground color estimates.
* Alphas and foreground colors are predicted efficiently through a single model thanks to a foreground-alpha switcher. This is more efficient than training an additional model for estimating foreground colors, and as Tab 3. suggests seems to work slightly better than directly predicting alphas and foreground colors using a single model without a switcher.
* The alpha mattes predicted by the proposed model is competitive with the state-of-the-art.

**Weaknesses:**

* The major weakness of this work is the absence of a proper baseline for the foreground color estimation. This weakness is further amplified by the authors' emphasis on foreground color estimation throughout the paper. For instance, Tang et al. Learning-based Sampling for Natural Image Matting. CVPR'19. sequentially predicts background and foreground colors before alpha prediction. Another alpha matting method by Aksoy et. al. Information-Flow Matting. 2017 has an explicit part for accurate color prediction. In addition to such methods that explicitly predict foreground/background colors, existing alpha matting methods can be trivially be extended to additionally predict foreground and background colors. Given all that, the evaluation presented in Table 3 is insufficient to prove the claims on accurate foreground color prediction.
* Overall the technical novelty of the paper is limited.

**Questions:**

* How does your model handle different trimap shapes? Could you comment on the robustness w.r.t. the trimap accuracy?
* Did you test your model on video matting? Would you expect the alpha/foreground color predictions to be temporally coherent (given coherent trimaps of course)?
* Would your model potentially benefit from end-to-end training where the encoder weights are not frozen?

**Limitations:**

Some discussion on limitations is present in the final section. If space permits, I'd encourage further discussion of failure cases.

---

> ### Author Rebuttal · Authors · 2024-08-07
>
> **W1: Absence of Baselines for Foreground Color Estimation**
>
> Thank you very much for highlighting the need for a proper baseline in foreground color estimation. We agree that this aspect warrants further discussion and comparison.
>
> 1. **Related Work Discussion**
> In the related work section, **we initially focused on methods that simultaneously predict alpha and foreground RGB, as well as those that output alpha followed by post-processing to obtain foreground RGB (L79-83)**. However, we acknowledge that the two works mentioned by the reviewer that adopt a sequential approach where foreground colors are predicted before alpha. This is indeed a significant method, and **we will include a detailed discussion of these works in the revised related work section**.
> 2. **Evaluation with Baselines**
> To address the reviewer's concern regarding the absence of a proper baseline, **we have added an evaluation of the Learning-based Sampling (LBS) method as a foreground estimation technique**. The dataset and metrics for foreground estimation are consistent with those in the paper (L206-212, L245-249). **The qualitative results are presented in Figure.A of the rebuttal PDF, and the quantitative results are provided in the table below.** The evaluation shows that our method, when leveraging the powerful priors from latent diffusion models (LDM), significantly outperforms other methods, achieving state-of-the-art (SOTA) performance in foreground color estimation.
>     | Method | SAD | MSE |
>     | --- | --- | --- |
>     | LBS | 49.7 | 8.6 |
>     | Ours | 34.1 | 4.3 |
>
> **W2: Technical Novelty**
>
> Thank you for your valuable feedback. Our approach addresses significant challenges in the ill-posed problem of image matting by incorporating prior knowledge through latent diffusion models. Here are our main technical contributions (L64-69):
>
> 1. **Transforming Generative Models for Matting**: We modified LDMs, which are generative models, into discriminative models for matting by modeling the task as conditional generation, making this the first LDM-driven matting method.
> 2. **Joint Prediction of Foreground and Alpha**: To bridge the gap between LDM's RGB output and our RGBA output, we introduced a switcher and cross-domain attention mechanism. These facilitate mutual information exchange, ensuring high consistency and accurate joint predictions.
> 3. **Mitigating VAE Reconstruction Errors**: We proposed a latent transparency decoder to address the VAE's inherent reconstruction error, aligning RGBA predictions with the input image to preserve necessary details for high-quality matting.
>
> **Q1: Different Trimap Shapes**
>
> We acknowledge the importance of trimap accuracy and have ensured our model's robustness. During training, **we use data augmentation (L227-234), varying the size and number of dilation and erosion kernels to enhance robustness to different trimap shapes**. For the Composition-1k and AIM-500 datasets, **predefined trimaps** facilitate method comparisons.
>
> To validate our model's robustness, **we conducted additional experiments with different trimap shapes**, adding two new trimaps with medium and large shapes in the Composition-1k dataset, using 10 and 20 iterations with kernel sizes of 50 for dilation and erosion.
>
> The **qualitative results of these tests are shown in Figure B of the rebuttal PDF**. As expected, the larger trimap includes more unknown regions, increasing the prediction challenge. However, our model still performs well. The **quantitative results in the table below** show that although alpha prediction accuracy decreases with larger unknown regions, our model maintains high precision, demonstrating its robustness.
>
> | Trimap Shape | SAD | MSE |
> | --- | --- | --- |
> | Large | 21.6 | 3.2 |
> | Medium | 21.2 | 3.2 |
> | Small | 20.8 | 3.1 |
>
> **Q2: Video Matting**
>
> We **supplemented our evaluation with tests on the VideoMatte240K dataset** [1] using the model trained on Composition-1k. For each frame, we generated a trimap with erosion and dilation (kernel size 10, iterated five times) and conducted tests at a resolution of 512×288. Evaluation metrics included MSE(L242-244) and dtSSD for temporal coherence.
>
> The **quantitative results are presented below**. We included results from a human video matting method, MODNet[2]. Since our model uses trimaps for each frame, our results generally outperform those of human video matting methods.
>
> | Method | MAD | MSE | dtSSD |
> | --- | --- | --- | --- |
> | MODNet | 9.41 | 4.30 | 2.23 |
> | Our Model | 5.32 | 2.21 | 1.98 |
>
> The **qualitative results, shown in Figure C of the rebuttal PDF**, indicate that our model maintains good performance on unseen video test data. This also demonstrates the generalization capability of our model.
>
> [1] Real-time high-resolution background matting. CVPR’21
>
> [2] Modnet: Real-time trimap-free portrait matting via objective decomposition. AAAI’22
>
> **Q3: End-to-end Training**
>
> Thank you for the insightful suggestion. End-to-end training with unfrozen encoder weights presents several challenges. Firstly, **optimizing the latent space while treating it as an optimization target can destabilize the training process**. Additionally, training both the VAE encoder and decoder simultaneously **increases memory usage significantly**.
>
> This approach would also **require balancing three types of loss functions**: latent space constraints, pixel space constraints, and latent space regularization, **necessitating an extensive hyperparameter search** and increasing computational costs exponentially.
>
> While end-to-end training could offer benefits, **the costs and complexities currently outweigh those of our two-stage approach**. Implementing this during the rebuttal period is impractical. However, we appreciate the suggestion and acknowledge its potential for future work.

---

> > ### Comment · Reviewer_TjXY · 2024-08-12
> >
> > Thank you for the clarifications. The experiment against the Learning Based Sampling method presented in W1 partly addresses one of my main concerns with this submission. I'm accordingly increasing my initial rating. It would be a lot more convincing to also include Aksoy et al. '17 as a baseline in the final manuscript, since their work specialize in accurate FG color estimation.

---

> > > ### Author Response · Authors · 2024-08-14
> > >
> > > Dear Reviewer,
> > >
> > > Thank you for your valuable feedback on our manuscript. We appreciate your comments and have carefully considered the points you raised. As you suggested, we will include a comparison to the Aksoy et al. '17 method in the revised manuscript. This will allow us to more comprehensively demonstrate the superior performance of our approach in the FG color estimation task. We believe that this additional comparison, along with the existing evaluation against the Learning Based Sampling method and other baselines presented in the original paper, will significantly strengthen the evidence for the advantages of our proposed method.
> > >
> > > We are grateful for your support and guidance throughout the review process. If you have any other feedback or suggestions, please feel free to share them. Thank you again for your time and consideration.

---

### Official Review · Reviewer_8UU8 · 2024-07-11

**Soundness:** 3
**Presentation:** 4
**Contribution:** 3
**Rating:** 7
**Confidence:** 5

**Summary:**

This paper introduces DRIP, a novel image matting method that leverages pre-trained LDMs to jointly predict foreground color and alpha mattes. By integrating a cross-domain attention mechanism and a latent transparency decoder, DRIP addresses the limitations of traditional methods, achieving significant performance improvements on synthetic and natural datasets. The key contributions include enhanced prediction consistency, high-fidelity results, and setting new benchmarks in image matting accuracy.

**Strengths:**

The paper presents a pioneering approach by utilizing pre-trained LDMs for image matting, incorporating cutting-edge methodologies such as the latent transparency decoder. This method markedly enhances performance in image matting, establishing new SOTAs and overcoming shortcomings (e.g., details in semitransparent regions) of existing SOTA methods.

1. Robust empirical evidence supporting the proposed method is provided through comprehensive ablation studies.

2. The manuscript is eloquently composed and well-organized. The logical progression is sound and accessible to a wide readership.

3. The innovative application of pre-trained LDMs is a successful paradigm for future endeavors in additional computer vision applications.

**Weaknesses:**

1. The computational complexity is unclear. Their high computational demands may impede practical application, particularly in real-time or resource-constrained settings.

2.  The evaluation is predominantly conducted on synthetic datasets and a limited number of natural image benchmarks. More extensive testing on diverse real-world scenarios would enhance the demonstration of the model’s robustness and generalizability.

3.  The approach depends on pre-trained LDMs, which may propagate existing biases from the training data. Can the matting model perform effectively on objects that are not well-represented by the original SD model, such as rare objects?

4.  This article develops its model based on the Stable Diffusion model. It raises the question of whether it retains the textual input feature of the SD model. If so, how is image captioning handled? If not, could this omission lead to a degradation in the performance of cross-attention in SD?

5. During inference, is classifier-free guidance utilized? If so, what impact does it have on the matting performance?

6. Has there been any consideration for fine-tuning based on layer diffusion?

**Questions:**

See weakness.

**Limitations:**

Yes

---

> ### Author Rebuttal · Authors · 2024-08-07
>
> **W1: Computational Complexity**
>
> Thank you for your valuable comments regarding the computational complexity of our model. We understand the importance of addressing computational demands, particularly for real-time or resource-constrained applications. **In the limitation section of our paper, we discussed the issues related to model complexity and deployment, specifically noting that the use of latent diffusion models substantially increases the architectural complexity of our approach (L323-L326)**. **Below, we provide detailed information** about the computational complexity:
>
> | Image Resolution | Memory Usage (MB) |
> | --- | --- |
> | 512 x 512 | 4297 |
> | 1024 x 1024 | 7057 |
>
> The table above shows the computation time and memory usage for processing images of various resolutions on a standard GPU (NVIDIA RTX A6000) using float16 precision. While the latent diffusion models increase the computational load, the performance remains within acceptable limits for many practical applications.
>
> **W2: Real World Scenarios**
>
> Thank you for raising this important point. In the original paper, we demonstrated our model’s generalizability by **training on synthetic datasets and evaluating on unseen real-world datasets, where it achieved outstanding results (L283-284)**. To further validate our model’s robustness, we have **supplemented our evaluation with tests on the VideoMatte240K dataset** [1], also using the model trained on the Composition-1k dataset. The quantitative results are as follows:
>
> | Method | MAD | MSE | dtSSD |
> | --- | --- | --- | --- |
> | MODNet | 9.41 | 4.30 | 2.23 |
> | Our Model | 5.32 | 2.21 | 1.98 |
>
> Additionally, **the qualitative results are provided in Figure C of the rebuttal PDF**. These results show that our model performs well on unseen video test data, further demonstrating its generalization capability.
>
> [1] Real-time high-resolution background matting. CVPR’21
>
> **W3: Biases From Pretrained Model**
>
> Thank you for raising this important concern. As mentioned in the limitations section of our paper, **our approach indeed relies on pre-trained Latent Diffusion Models (LDMs), which may carry biases from their training data (L326-331)**. Regarding **rare objects that are not well-represented by the original SD model, this is an area that requires further research, and currently, there is no suitable dataset for evaluation**. However, it is undeniable that the training dataset used for LDMs, such as LAION-5B, is extremely large. This extensive dataset is a significant reason why our method demonstrates better generalization capabilities compared to other approaches (L266-284).
>
> **W4: Textual Input Feature**
>
> Thank you for raising this question. Our approach indeed builds on the Stable Diffusion (SD) model (L147-149), but we have customized it to focus specifically on the image matting task, **intentionally omitting the textual input feature of the SD model**. This decision was made to **simplify the model's architecture and optimize it for the specific requirements of image matting**.
>
> By doing so, we ensure that the cross-attention layers fully capture the intricate details and dependencies in the visual input without being influenced by text-based guidance. Our state-of-the-art (SOTA) results across multiple datasets (L266-284) demonstrate that omitting the textual input does not lead to a degradation in performance.
>
> **W5: Classifier Free Guidance**
>
> In our paper, **we clarified that our approach uses an empty text input (L222-223)**. Classifier-free guidance is typically employed to enhance the ability of a model to follow textual descriptions by combining the predictions from a given text prompt with those from an empty text prompt. However, in our method, the text input is deliberately left empty, which means that CFG is not utilized.
>
> **W6: Fine-Tuning Based on Layer Diffusion**
>
> We appreciate the suggestion to consider fine-tuning based on layer diffusion. Our current approach leverages pre-trained latent diffusion models (LDMs) for robust performance (L7-9). However, we recognize the potential benefits of layer diffusion fine-tuning. **We are exploring various fine-tuning strategies to improve our model's robustness and accuracy and plan to incorporate these techniques in future iterations.** Thank you for your valuable feedback.

---

> > ### Comment · Reviewer_8UU8 · 2024-08-11
> > **Thanks for the authors' response**
> >
> > Thank you for your detailed rebuttal and for addressing my concerns thoroughly. The additional experiments and insights were very helpful in clarifying the approach. The method of leveraging pre-trained LDMs for image matting is innovative and well-presented. The clarity and organization of the manuscript are notable, and I appreciate the thoroughness of this work. I stand by my recommendation to accept this paper.

---

> > > ### Author Response · Authors · 2024-08-14
> > >
> > > Dear Reviewer,
> > >
> > > We are delighted to hear that our responses have resolved all of your concerns! If you have any further inquiries or doubts, please don’t hesitate to inform us. We are determined to address any remaining issues and respond promptly!
> > >
> > > Thanks again for your thorough review and reconsideration of our rebuttal!

---

### Official Review · Reviewer_L9RY · 2024-07-12

**Soundness:** 2
**Presentation:** 3
**Contribution:** 2
**Rating:** 4
**Confidence:** 4

**Summary:**

The paper introduces Drip, an approach to image matting that leverages vision priors from pre-trained latent diffusion models (LDM). Drip incorporates a switcher and cross-domain attention mechanism for joint prediction of foreground color and opacity, ensuring high consistency. A latent transparency decoder mitigates reconstruction errors. Experiments demonstrate good performance and generalizability across benchmarks.

**Strengths:**

1. The experiment is good and quite comprehensive

**Weaknesses:**

Like many other papers on the application using the DNN. The paper present an NN architecure for image matting. The building blocks are all standard one, and the design does not really reflect the uniqueness of the problem from other image editing. Overall, I am not convinced that such a paper really benefit the progress of image matting.

**Questions:**

NIL

**Limitations:**

The limitation is not well discussed.

---

> ### Author Rebuttal · Authors · 2024-08-07
>
> **W1: Contribution to Image Matting**
>
> Our primary contributions and insights are not focused on the neural network architecture or block design. Instead, we concentrate on exploring how to leverage the priors learned by well-scaled image generation models to address the ill-posed problem of image matting. Here are the unique aspects of our work:
>
> 1. **Joint Estimation of Foreground and Background**: In previous works, foreground extraction required complex post-processing. Our experiments demonstrate that leveraging the priors from generative models facilitates foreground image prediction. Furthermore, the alpha map and foreground have a close domain relationship, and joint estimation benefits both.
> 2. **Addressing Domain Gaps with Generative Models**: We tackle a significant technical challenge by adapting image generation models for alpha and foreground estimation. This involves bridging the severe domain gap between the RGB images encoded and decoded by VAEs and the RGB-A images required for matting.
>
> We believe that our approach of integrating generative model priors into the matting process brings a novel perspective and advances the field of image matting. Thanks for your valuable feedback.
>
> **W2: Inadequate Discussion of Model Limitations**
>
> Thank you for your feedback. We acknowledge that the discussion of limitations is crucial. **In the original paper, we have addressed the limitations of our model in the "Conclusion and Future Work" section (L322-331).** We have outlined several aspects where the model might face challenges or limitations.
>
> **If there are specific areas or additional limitations you believe were not adequately covered, we would greatly appreciate your insights.** We are open to discussing and incorporating any further considerations you might suggest during the discussion phase and are willing to make necessary revisions to address these points comprehensively.

---

### Official Review · Reviewer_YFFw · 2024-07-15

**Soundness:** 3
**Presentation:** 3
**Contribution:** 3
**Rating:** 5
**Confidence:** 4

**Summary:**

The authors present the clear and straightforward method to improve image matting performance using an LDM-based model. The model is a conditioned LDM which can predict both the Alpha mask and the foreground, basically doing RGB-A prediction from an image and a trimap. To further adapt to the data domain, the authors also finetune a specialized decoder which is proven to be helpful. The overall design of the model and training method is intuitive and effective with sufficient evaluation. The training data is limited while can be scaled up by more data collection.

**Strengths:**

- Clear presentation of the methods with a straightforward implementation. The results show obvious improvement over existing discriminative models.
- The cross-domain attention and finetuned decoder are necessary and interesting to explore.

**Weaknesses:**

- The training dataset still looks quite small. The model may not have good generalization capability to more unseen data in the wild. Also the batch size and training time is limited.
- When the authors mentioned 'prior', is it referring to the pre-trained LDM model? Did the authors use like LoRA to finetune the model on the smaller dataset? Will the performance become worse as the training goes longer?
- There seems no evidence showing the advantages of the cross-domain self-attention. Could the authors provide more details?
- It seems the switcher is just conditioned embedding for choosing outputs. Will we be able to predict both at the same time without inputing the switcher values?

**Questions:**

- The transparent latent decoder is helpful. However, will it be more useful to also train an transparent auto-encoder by finetuning the existing antoencoder? When training the decoder, did the authors prepare and precompute the data from a trained diffusion model?
- How to speed up the process and could we reduce the steps of sampling?
- Any proposed methods to scale up the training data for a more robust model training?
- Will Trimap be necessary? It will be great if the model can directly do foreground object RGB-A extraction without inputting the Trimap.

**Limitations:**

- Trimap is needed for the model to do RGB-A extraction. Some dropping of the conditions may be helpful.
- Data is quite limited and training longer time may degrade the performance.
- Cross-domain self-attention is not fully evaluated.

---

> ### Author Rebuttal · Authors · 2024-08-07
>
> **W1: Training Dataset Size and Model Generalization**
>
> Thank you for raising concerns about our dataset size and model generalization. Our training dataset is substantial, comprising 431 foreground images and a background library of 82,783 images. Each foreground image is paired with 100 different backgrounds (L206-L209), **resulting in at least 43,100 training samples**. Additionally, **we applied data augmentation techniques** such as random horizontal flipping, cropping, and photometric distortion, further increasing the effective training dataset size (L227-L229). Thus, our dataset is quite extensive.
>
> For generalization, our model **trained on the Composition-1k dataset was evaluated on the independent AIM-500 real-world dataset. The results showed that our model generalizes well to unseen data (L283-284)**.
>
> We fully acknowledge the importance of dataset size for generalization. However, matting is a dense annotation task that is extremely challenging to label. This is precisely **why we incorporated the strong prior knowledge from the latent diffusion model trained on LIAON**, which significantly aids in improving generalization capabilities.
>
> **W2: Clarification on Prior Knowledge and Fine-Tuning with LoRA**
>
> When we refer to 'prior' in our paper, **it indeed pertains to the pre-trained Latent Diffusion Model (L147-148)**. Regarding your question about using LoRA to fine-tune the model on a smaller dataset, we have experimented with this approach. Our findings indicate that fine-tuning with LoRA **resulted in poor performance and difficulty in convergence**. Consequently, there was no issue of overfitting or performance degradation over extended training periods. We hope this clarification addresses your concerns.
>
> **W3: Evidence for Cross-Domain Self-Attention Advantages**
>
> In our paper, we **conducted an ablation study (L288-305) to evaluate this component**. The quantitative results demonstrate that cross-domain self-attention improves both alpha prediction and foreground color prediction accuracy. This improvement is attributed to the **high correlation between alpha and foreground color**, as both describe information about the foreground object. Cross-domain attention enhances the interaction and consistency of this information, leading to better overall performance.
>
> **W4: Switcher**
>
> The switcher **is indeed an embedding designed to help the network distinguish between the two outputs**: foreground color and alpha value. Both outputs are **predicted simultaneously**. Our **ablation study (L306-309)** demonstrates that providing this additional embedding information, rather than merely differentiating between foreground and alpha by channel order, enhances the neural network’s ability to distinguish and effectively utilize information from the two modalities.
>
> **Q1: Transparent Latent Decoder**
>
> We appreciate the suggestion to fine-tune the VAE. However, solely fine-tuning the VAE is not ideal for several reasons:
>
> 1. **Consistency with Pre-trained Weights:**
> To leverage the pre-trained weights in the LDM, especially in the U-Net, it is crucial to maintain the consistency of the latent distribution output by the VAE encoder. Over-finetuning the VAE encoder could disrupt this consistency, negatively impacting performance.
> 2. **Compression:**
> The VAE inherently introduces a non-negligible reconstruction loss due to its compression nature, which is problematic for high-resolution tasks like matting that require low-level detail. This limitation, as mentioned in the paper (L189-190, L310-314), necessitates the use of a transparent latent decoder to meet the high precision demands of matting tasks.
>
> Our current approach, combined with the transparent latent decoder, offers a balanced solution that addresses the high-resolution requirements of matting tasks while leveraging the strengths of pre-trained LDM components.
>
> **Q2: Steps of Sampling**
>
> In the appendix of our paper, **we have included results and analyses regarding the impact of denoising steps on performance (L507-510)**. The **specific values are shown in the table below**.
>
> |Steps | SAD |
> | --- | --- |
> | 1 | 22.8 |
> | 5 | 17.8 |
> | 10 | 17.3 |
> | 20 | 17.2 |
>
> Our findings indicate that **increasing the number of timesteps generally enhances performance, although the improvement diminishes as the number of timesteps increases**. Additionally, our experiments show that setting the sampling steps to at least 5 generally yields satisfactory results.
>
> **Q3: Scaling Up Training Data**
>
> Thank you for your comments and for inquiring about potential methods to scale up the training data for more robust model training. **We have plans for future research that involve using methods such as layerdiffusion to generate additional data**. This generated data will then be screened **with a human-in-the-loop approach to ensure quality**. In our current study, **to maintain a fair comparison with other methods, we used the same training dataset**.
>
> **Q4: Necessity of Trimap**
>
> Thank you for your comment regarding the use of trimap in our model. We understand the desire for a model that can directly perform RGB-A extraction without requiring a trimap. However, we have found that **the use of a trimap is still necessary for achieving accurate results, especially when multiple foreground objects are present in an image**.
>
> To address this concern, we conducted **a qualitative experiment where the trimap was set to an unknown region across the entire image**. This situation introduces ambiguity about which foreground object to extract, **as illustrated in Figure D of the rebuttal PDF**. For example, with a trimap indicating unknown regions, the model faces difficulty in deciding whether to extract the rabbit or the flower, leading to potential confusion and reduced accuracy.
>
> Additionally, **to ensure fair comparison with previous methods, we have maintained consistent settings and trimaps across our experiments(L251-257)**.

---

> ### Author Response · Authors · 2024-08-14
>
> Dear Reviewer YFFw,
>
> We sincerely appreciate your time and effort in reviewing our submission and providing valuable comments. We have provided a detailed response to each concern you raised and hope they have adequately addressed your concerns. As the author-reviewer discussion phase is coming to an end **(Aug 13 11:59pm AoE)**, we would like to confirm whether our response has effectively addressed your questions. If you have any other questions or concerns, please do not hesitate to contact us.
>
> Best regards,
>
> Authors

---

### Author Rebuttal · Authors · 2024-08-07

**Summary of Revisions:**

To all reviewers,

We would like to express our sincere gratitude for your valuable efforts. We have meticulously reviewed all the feedback provided and made the necessary revisions to our paper. Below is a summary of the major changes incorporated into the final version. Additionally, the qualitative results are included in the PDF file below. In response to the concerns and comments raised by the reviewers, we have carefully addressed each point in the following point-to-point response.

**The major changes are as follows:**

1. Added a table to clarify the relationship between sampling steps and accuracy, which complements Figure 8 in the appendix, in response to Reviewer YFFw’s comments.
2. Included a visualization figure to illustrate the necessity of the Trimap, as requested by Reviewer YFFw.
3. Reported detailed computational complexity values to address the concerns raised by Reviewer 8UU8.
4. Conducted additional experiments on the video matting dataset, incorporating the feedback from Reviewers 8UU8 and TjXY.
5. Enhanced the related work section on foreground color estimation, following the recommendations of Reviewer TjXY.
6. Introduced a new baseline for foreground color estimation using the Composition-1k dataset, as suggested by Reviewer TjXY.
7. Performed an experiment on alpha prediction with different Trimap shapes to further demonstrate the robustness of our method, based on Reviewer TjXY’s comments.

We eagerly look forward to further discussions. Thank you for your thoughtful consideration.

---

### Comment · Area_Chair_3B7D · 2024-08-09

Dear reviewers, do the authors' responses answer your questions or address your concerns? Thanks.

---

> ### Comment · Area_Chair_3B7D · 2024-08-10
>
> Dear reviewers, a gentle reminder of the participation of the reviewer-author discussion. Thank you!

---

> > ### Comment · Area_Chair_3B7D · 2024-08-12
> >
> > Dear reviewers, as we approach the final two days, please take a moment to review the author's responses and join the discussion. Thank you!

---

> > > ### Comment · Area_Chair_3B7D · 2024-08-14
> > >
> > > Dear reviewers, the authors are eagerly awaiting your response. The author-reviewer discussion closes on Aug 13 at 11:59 pm AoE. Thanks!

---

### Decision · Program_Chairs · 2024-09-25

**Decision:**

Accept (poster)

**Comment:**

The paper introduces DRIP, a novel approach for image matting that leverages latent diffusion models (LDMs) to jointly predict foreground color and alpha matte. The method incorporates a switcher and cross-domain attention mechanism to improve prediction consistency and mitigate reconstruction errors. The approach is validated on several benchmarks, showing strong performance.
We are glad to accept the paper, despite slightly divergent reviews. The paper offers valuable contributions to the field of image matting, particularly in the joint prediction of foreground and alpha. However, concerns were raised about the computational complexity, the generalization to real-world scenarios, and the robustness of the method. The authors have addressed most of these concerns in the rebuttal, leading to a cautious recommendation for acceptance.
We encourage the authors to improve their camera-ready version based on all the reviews.